# Towards Binary-Valued Gates
# for Robust LSTM Training

## Abstract

Long Short-Term Memory (LSTM) is one of the most widely used recurrent structures in sequence modeling. Its goal is to use gates to control the information flow (e.g., whether to skip some information/transformation or not) in the recurrent computations, although its practical implementation based on soft gates only partially achieves this goal and is easy to overfit. In this paper, we propose a new way for LSTM training, which pushes the values of the gates towards 0 or 1. By doing so, we can (1) better control the information flow: the gates are mostly open or closed, instead of in a middle state; and (2) avoid overfitting to certain extent: the gates operate at their flat regions, which is shown to correspond to better generalization ability. However, learning towards discrete values of the gates is generally difficult. To tackle this challenge, we leverage the recently developed Gumbel-Softmax trick from the field of variational methods, and make the model trainable with standard backpropagation. Experimental results on language modeling and machine translation show that (1) the values of the gates generated by our method are more reasonable and intuitively interpretable, and (2) our proposed method generalizes better and achieves better accuracy on test sets in all tasks. Moreover, the learnt models are not sensitive to low-precision approximation and low-rank approximation of the gate parameters due to the flat loss surface.

## 1 Introduction

Recurrent neural networks (RNN) (Hochreiter, 1998) are widely used in sequence modeling tasks, such as language modeling (Kim et al., 2016; Jozefowicz et al., 2016), speech recognition (Zhang et al., 2016), time series prediction (Xingjian et al., 2015), machine translation (Wu et al., 2016; Britz et al., 2017), image captioning (Vinyals et al., 2015; Xu et al., 2015), and image generation (Villegas et al., 2017).

To address the long-term dependency and gradient vanishing problem of conventional RNN, long short-term memory (LSTM) (Gers et al., 1999; Hochreiter & Schmidhuber, 1997b) was proposed, which introduces *gate functions* to control the information in a recurrent unit: a *forget gate function* to determine how much previous information should be excluded for the current step, an *input gate function* to find relevant signals to be absorbed into the hidden context, and an *output gate function* for prediction and decision making. For ease of optimization, in practical implementation, one usually uses element-wise sigmoid function to mimic the gates, whose outputs are soft values between 0 and 1. By using such gates, LSTM usually performs much better than conventional RNN. However, the benefits come with the cost of introducing many more parameters in the gates, which makes the training of a LSTM model inefficient and easy to overfit (Krueger et al., 2016; Zaremba et al., 2014; Semeniuta et al., 2016).

In this paper, we explore a new way to train LSTM by pushing the values of the gates to the boundary of their ranges $(0, 1)$ [1]. Pushing the values of the gates to 0/1 has certain advantages. First, it well aligns with the original purpose of the development of gates: to get the information in or skip by "opening" or "closing" the gates during the recurrent computation. Second, training LSTM

---

[1] The output of a gate function is usually a vector. For simplicity, in the paper, we say "pushing the output of the gate function to 0/1" when meaning "pushing each dimension of the output vector of the gate function to either 0 or 1". We also say that each dimension of the output vector of the gate function is a gate, and say a gate is open/closed if its value is close to 1/0.

towards binary-valued gates can make the learnt model generalize better. According to (Hochreiter & Schmidhuber, 1997a; Haussler et al., 1997; Keskar et al., 2016; Chaudhari et al., 2016), a model lying in a flat region of the loss surface is likely to generalize well, since any small perturbation to the model makes little fluctuation to the loss. Training LSTM towards binary-valued gates means seeking a set of parameters to make the values of the gates approaching zero or one, namely residing in the flat region of the sigmoid function. Simple deductions show that this also corresponds to the flat region of the overall loss surface.

Technically, pushing the outputs of the gates towards such discrete values is challenging. A straightforward approach is to sharpen the sigmoid function by a smaller temperature. However, this is equivalent to rescaling the input and cannot guarantee the values of the learnt gates to be close to 0 or 1. To tackle this challenge, in this paper, we leverage the Gumbel-Softmax trick that Jang et al. (2016) and Maddison et al. (2016) recently develop for variational methods. The trick aims to generate approximated samples for categorical latent variables in a stochastic computational graph, e.g., variational autoencoder, brings convenience to using reparametrization tricks, and thus leads to efficient learning. Specifically, during training, we apply the Gumbel-Softmax trick to the gates to approximate the values sampled from the Bernoulli distribution given by the parameters, and train the LSTM model with standard backpropagation methods. We call this method Gumbel-Gate LSTM ($G^2$-LSTM). We conduct three experiments on two tasks (language modeling and machine translation) to verify our proposed method. We have the following observations from experimental results:

- Our model generalizes well: In all tasks, we achieve superior performance to baseline algorithms on the test sets, and the gap between training and test is effectively reduced.
- Our model is not sensitive due to its flat loss surface: We apply several model compression algorithms to the parameters in the gates, including low-precision approximation and low-rank approximation, and all results show that our learnt models are better.
- The gates in our learnt model are meaningful and intuitively interpretable after visualization. Furthermore, our model can automatically learn the boundaries inside the sentences.

The organization of the paper is as follows. We introduce related work in Section 2 and propose our learning algorithm in Section 3. Experiments are reported in Section 4 and future work is discussed in the last section.

## 2 RELATED WORK

### 2.1 LOSS SURFACE AND GENERALIZATION

The concept of sharp and flat minima has been first discussed in (Hochreiter & Schmidhuber, 1997a; Haussler et al., 1997) . Intuitively, a flat minimum $x$ of a loss $f(\cdot)$ corresponds to the point for which the function $f$ varies slowly in a relatively large neighborhood of $x$. In contrast, a sharp minimum $x$ is such that the function $f$ increases rapidly in a small neighborhood of $x$. The sensitivity of the loss function at sharp minima negatively impacts the generalization ability of a trained model on new data. Recently, several papers discuss how to modify the training process and to learn a model in a flat region so as to obtain better generalization ability. Keskar et al. (2016) show by using small-batch training, the learnt model is more likely to converge to a flat region rather than a sharp one. Chaudhari et al. (2016) propose a new objective function considering the local entropy and push the model to be optimized towards a wide valley.

### 2.2 DROPOUT IN RECURRENT NEURAL NETWORK

Dropout is one of the most standard tricks used in deep learning to improve generalization ability. For recurrent neural networks, Zaremba et al. (2014) and Semeniuta et al. (2016) apply dropout to feed-forward connections and recurrent units of RNNs. In Zoneout (Krueger et al., 2016), the values of the hidden states and memory cells are randomly either maintained by their previous value or updated as usual, which introduces stochastic identity connections between subsequent time steps.

Different from dropout, which is to regularize the training of a deep neural network by randomly dropping nodes/edges to prevent co-adaptations, our method is to bias the optimization process and

ensure to find a model in a flat region to avoid overfitting. Therefore, our method is complementary to dropout in RNNs, and actually in our experiments our method is well combined with dropout.

## 2.3 GUMBEL-SOFTMAX TRICK

Jang et al. (2016) and Maddison et al. (2016) develop a continuous relaxation of discrete random variables in stochastic computational graphs. The main idea of the method is that the multinomial distribution can be represented according to Gumbel-Max trick, thus can be approximated by Gumbel-Softmax distribution. In detail, given a probability distribution over $k$ categories with parameter $\pi_1, \pi_2, \ldots, \pi_k$, the Gumbel-Softmax trick approximately samples the categorical variable according to:

$$y_i = \frac{\exp((\log \pi_i + q_i)/\tau)}{\sum_{j=1}^{k} \exp((\log \pi_j + q_j)/\tau)} \qquad \text{for } i = 1, \ldots, k, \tag{1}$$

where $\tau$ is the temperature and $q_i$ is independently sampled from Gumbel distribution: $q_i = -\log(-\log U_i), U_i \sim \text{Uniform}(0, 1)$.

By using the Gumbel-Softmax trick, we can generate sample $y = (y_1, ..., y_k)$ to approximate the categorical distribution. Furthermore, as the randomness $q$ is independent of $\pi$ (which is usually defined by a set of parameters), we can use reparameterization trick to optimize the model parameters using standard backpropagation algorithms. Gumbel-Softmax trick has been adopted in several applications such as variation autoencoder (Jang et al., 2016), generative adversarial net (Kusner & Hernández-Lobato, 2016), and language generation (Subramanian et al., 2017). To the best of our knowledge, this is the first work to introduce the Gumbel-Softmax trick in LSTM for robust training purpose.

## 3 THE PROPOSED TRAINING ALGORITHM

In this section, we present a new and robust training algorithm for LSTM by learning towards binary-valued gates.

### 3.1 BACKGROUND

Recurrent neural networks process an input sequence $\{x_1, x_2, \ldots, x_T\}$ sequentially and construct a corresponding sequence of hidden states/representations $\{h_1, h_2, \ldots, h_T\}$. In single-layer recurrent neural networks, the hidden states $\{h_1, h_2, \ldots, h_T\}$ are used for prediction or decision making. In deep (stacked) recurrent neural networks, the hidden states in layer $k$ are used as inputs to layer $k + 1$.

In recurrent neural networks, each hidden state is trained (implicitly) to remember and emphasize task-relevant aspects of the preceding inputs, and to incorporate new inputs via a recurrent operator, $T$, which converts the previous hidden state and presents input into a new hidden state, e.g.,

$$h_t = T(h_{t-1}, x_t) = \tanh(W_h h_{t-1} + W_x x_t + b),$$

where $W_h$, $W_x$ and $b$ are parameters.

Long short-term memory RNN (LSTM) (Hochreiter & Schmidhuber, 1997b) is a carefully designed recurrent structure. In addition to the hidden state $h_t$ used as a transient representation of state at timestep $t$, LSTM introduces a memory cell $c_t$, intended for internal long-term storage. $c_t$ and $h_t$ are computed via three gate functions. The forget gate function $f_t$ directly connects $c_t$ to the memory cell $c_{t-1}$ of the previous timestep via an element-wise multiplication. Large values of the forget gates cause the cell to remember most (if not all) of its previous values. The other gates control the flow of information in input ($i_t$) and output ($o_t$) of the cell. Each gate function has a weight matrix and a bias vector; we use subscripts $f$, $i$ and $o$ to denote parameters for the forget gate function, the input gate function and the output gate function respectively, e.g., the parameters for the forget gate function are denoted by $W_{xf}, W_{hf}$, and $b_f$.

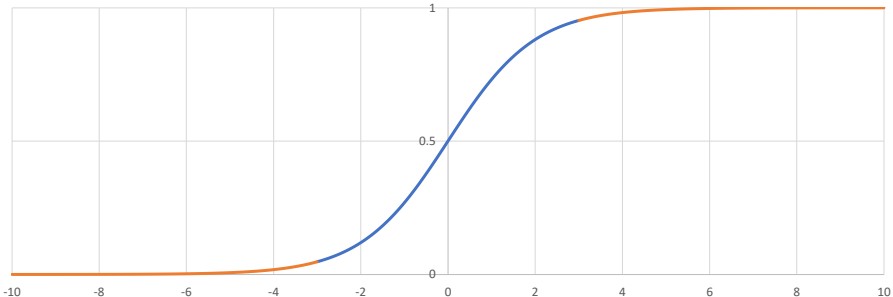

Figure 1: The orange parts correspond to the saturation area of the sigmoid function.

With the above notations, an LSTM is formally defined as follows:

$$
\begin{aligned}
i_t &= \sigma(W_{xi}x_t + W_{hi}h_{t-1} + b_i) & (2) \\
f_t &= \sigma(W_{xf}x_t + W_{hf}h_{t-1} + b_f) & (3) \\
o_t &= \sigma(W_{xo}x_t + W_{ho}h_{t-1} + b_o) & (4) \\
g_t &= \tanh(W_{xg}x_t + W_{hg}h_{t-1} + b_g) & (5) \\
c_t &= f_t \odot c_{t-1} + i_t \odot g_t & (6) \\
h_t &= o_t \odot \tanh(c_t), & (7)
\end{aligned}
$$

where $\sigma(\cdot)$ represents the sigmoid function and $\odot$ is the element-wise product.

## 3.2 TRAINING LSTM GATES TOWARDS BINARY VALUES

The LSTM unit requires much more parameters than the simple RNN unit, and makes it hard to generalize. As we can see from Eqn (2) - (7), a large percentage of the parameters are used to compute the gate (sigmoid) functions. If we can push the outputs of the gates to the saturation area of the sigmoid function (i.e., towards 0 or 1), the loss function with respect to the parameters in the gates will be flat: if the parameters in the gates perturb, the change to the output of the gates is small due to the sigmoid operator (see Figure 1), and then the change to the loss is little, which means the flat region of the loss. As discussed in (Chaudhari et al., 2016), minima in a flat region is more likely to generalize better, and thus toward binary-valued gates will lead to better generalization.

However, the task of training towards binary-valued gates is quite challenging. One straightforward idea is to sharpen sigmoid function by using a smaller temperature, i.e., $f_{W,b}(x) = \sigma((Wx+b)/\tau)$, where $\tau < 1$ is the temperature. However, it is computationally equivalent to $f_{W',b'}(x) = \sigma(W'x + b')$ by setting $W' = W/\tau$ and $b' = b/\tau$. Then using a small temperature is equivalent to rescale the initial parameters as well as the gradients to a larger range. Usually, using an initial point in a large range with a large learning rate will harm the optimization process, and apparently cannot guarantee the outputs to be close to the boundary after training.

In this work, we leverage the recently developed Gumbel-Softmax trick. This trick is efficient in approximating discrete distributions, and is one of the widely used methods to learn discrete random variables in stochastic computational graphs. We first provide a proposition about the approximation ability of this trick for Bernoulli distribution, which will be used in our proposed algorithm.

**Proposition 1.** *Assume $\sigma(\cdot)$ is the sigmoid function. Given $\alpha \in \mathbb{R}$ and temperature $\tau > 0$, we define random variable $D_\alpha \sim B(\sigma(\alpha))$ where $B(\sigma(\alpha))$ is the Bernoulli distribution with parameter $\sigma(\alpha)$, and define $G(\alpha, \tau) = \sigma(\frac{\alpha + \log U - \log(1-U)}{\tau})$ where $U \sim Uniform(0,1)$. Then the following inequalities hold for arbitrary $\epsilon \in (0, \frac{1}{2})$,*

$$
P(D_\alpha = 1) - \frac{\tau}{4}\log(\frac{1}{\epsilon}) \leq \quad P(G(\alpha, \tau) \geq 1 - \epsilon) \quad \leq P(D_\alpha = 1), \tag{8}
$$

$$
P(D_\alpha = 0) - \frac{\tau}{4}\log(\frac{1}{\epsilon}) \leq \quad P(G(\alpha, \tau) \leq \epsilon) \quad \leq P(D_\alpha = 0). \tag{9}
$$

*Proof.* Since $\sigma^{-1}(x) = \log(\frac{x}{1-x})$, we have $P(G(\alpha, \tau) \geq 1 - \epsilon) = P(\frac{\alpha + \log U - \log(1-U)}{\tau} \geq \log(\frac{1}{\epsilon} - 1)) = P(e^{\alpha - \tau \log(\frac{1}{\epsilon} - 1)} \geq \frac{1-U}{U}) = P(U \geq \frac{1}{1 + e^{\alpha - \tau \log(\frac{1}{\epsilon} - 1)}}) = \sigma(\alpha - \tau \log(\frac{1}{\epsilon} - 1))$. Considering that sigmoid function is monotonically increasing and $\frac{1}{4}$-Lipschitz continuous, we have $P(D_\alpha = 1) - P(G(\alpha, \tau) \geq 1 - \epsilon) = \sigma(\alpha) - \sigma(\alpha - \tau \log(\frac{1}{\epsilon} - 1)) \geq 0$ and $P(D_\alpha = 1) - P(G(\alpha, \tau) \geq 1 - \epsilon) = \sigma(\alpha) - \sigma(\alpha - \tau \log(\frac{1}{\epsilon} - 1)) \leq \frac{\tau}{4} \log(\frac{1}{\epsilon} - 1) \leq \frac{\tau}{4} \log(\frac{1}{\epsilon})$. We omit the proof for (9) as it is almost identical to the proof of (8). □

We can see from the above proposition, the distribution of $G(\alpha, \tau)$ can be considered as an approximation of Bernoulli distribution $B(\sigma(\alpha))$. The rate of convergence is characterized by (8) and (9). When the temperature $\tau$ approaches positive zero, we directly obtain the following property which is also proved by Maddison et al. (2016),

$$P(\lim_{\tau \to 0^+} G(\alpha, \tau) = 1) = P(D_\alpha = 1), P(\lim_{\tau \to 0^+} G(\alpha, \tau)) = P(D_\alpha = 0). \tag{10}$$

We apply this method into the computation of the gates. Imagine an one-dimensional gate $\sigma(\alpha(\theta))$ where $\alpha$ is a scalar parameterized by $\theta$, and assume the model will produce a larger loss if the output of the gate is close to one, and produce a smaller loss if the gate value is close to zero. If we can repeatedly sample the output of the gate using $G(\alpha(\theta), \tau) = \sigma(\frac{\alpha(\theta) + \log U - \log(1-U)}{\tau})$ and estimate the loss, any gradient-based algorithm will push the parameter $\theta$ such that the output value of the gate is close to zero in order to minimize the expected loss. By this way, we can optimize towards the binary-valued gates.

As the gate function is usually a vector-valued function, we extend the notations into a general form: Given $\alpha \in \mathbb{R}^d$ and $\tau > 0$, we define $G(\alpha, \tau) = \sigma(\frac{\alpha + \log U - \log(1-U)}{\tau})$, where $U$ is a vector and each element $u_i$ in $U$ is independently sampled from Uniform$(0, 1)$, $i = 1, 2, \ldots, d$. In particular, we only push the outputs of input gates and forget gates towards binary values as the output gates usually need fine-granularity information for decision making which makes binary values less desirable (to justify this, we conducted similar experiments and observed a performance drop when pushing the output gates to 0/1 together with the input gates and the forget gates).

We call our proposed learning method Gumbel-Gate LSTM ($G^2$-LSTM), which works as follows during training:

$$
\begin{aligned}
i_t &= G(W_{xi}x_t + W_{hi}h_{t-1} + b_i, \tau) & (11) \\
f_t &= G(W_{xf}x_t + W_{hf}h_{t-1} + b_f, \tau) & (12) \\
o_t &= \sigma(W_{xo}x_t + W_{ho}h_{t-1} + b_o) & (13) \\
g_t &= \tanh(W_{xg}x_t + W_{hg}h_{t-1} + b_g) & (14) \\
c_t &= f_t \odot c_{t-1} + i_t \odot g_t & (15) \\
h_t &= o_t \odot \tanh(c_t). & (16)
\end{aligned}
$$

In the forward pass, we first independently sample values for $U$ in each time step, then update LSTM units using Eqn (11) - (16) and calculate the loss, e.g., negative log likelihood loss. In the backward pass, as $G$ is continuous and differentiable with respect to the parameters and the loss is continuous and differentiable with respect to $G$, we can use any standard gradient-based method to update the model parameters.

## 4 EXPERIMENTS

### 4.1 SETTINGS

We tested the proposed training algorithm on two tasks – language modeling and machine translation.

#### 4.1.1 LANGUAGE MODELING

Language modeling is a very basic task for LSTM. We used the Penn Treebank corpus which contains about 1 million words. The task is to train an LSTM model to correctly predict the next word

Table 1: Performance comparison on language model (perplexity)

| Model | Size | Valid | Test |
|---|---|---|---|
| *Existing results* | | | |
| Unregularzed LSTM | 7M | 120.7 | 114.5 |
| NR-dropout (Zaremba et al., 2014) | 66M | 82.2 | 78.4 |
| Zoneout (Krueger et al., 2016) | 66M | - | 77.4 |
| Variational LSTM (Gal & Ghahramani, 2016) | 19M | - | 73.4 |
| CharCNN (Kim et al., 2016) | 21M | 72.4 | 78.9 |
| Pointer Sentinel-LSTM (Merity et al., 2016) | 51M | - | 70.9 |
| LSTM + continuous cache pointer (Grave et al., 2016) | - | - | 72.1 |
| Variational LSTM + augmented loss (Inan et al., 2016) | 51M | 71.1 | 68.5 |
| Variational RHN (Zilly et al., 2016) | 23M | 67.9 | 65.4 |
| NAS Cell (Zoph & Le, 2016) | 54M | - | 62.4 |
| 4-layer skip connection LSTM (Melis et al., 2017) | 24M | 60.9 | 58.3 |
| AWD-LSTM w/o finetune (Merity et al., 2017) | 24M | 60.7 | 58.8 |
| AWD-LSTM (Baseline) (Merity et al., 2017) | 24M | 60.0 | 57.3 |
| *Our system* | | | |
| Sharpened Sigmoid AWD-LSTM w/o finetune | 24M | 61.6 | 59.4 |
| Sharpened Sigmoid AWD-LSTM | 24M | 59.9 | 57.5 |
| $G^2$-LSTM w/o finetune | 24M | **60.4** | **58.2** |
| $G^2$-LSTM | 24M | **58.5** | **56.1** |
| *+continuous cache pointer* | | | |
| AWD-LSTM + continuous cache pointer (Merity et al., 2017) | 24M | 53.9 | 52.8 |
| Sharpened Sigmoid AWD-LSTM + continuous cache pointer | 24M | 53.9 | 53.2 |
| $G^2$-LSTM + continuous cache pointer | 24M | **52.9** | **52.1** |

Table 2: Performance comparison on machine translation (BLEU)

| English→German task | BLEU | German→English task | BLEU |
|---|---|---|---|
| *Existing end-to-end system* | | | |
| RNNSearch-LV (Jean et al., 2015) | 19.40 | BSO (Wiseman & Rush, 2016b) | 26.36 |
| MRT (Shen et al., 2015) | 20.45 | NMPT (Huang et al.) | 28.96 |
| Global-att (Luong et al., 2015) | 20.90 | NMPT+LM (Huang et al.) | 29.16 |
| GNMT (Wu et al., 2016) | **24.61** | ActorCritic (Bahdanau et al., 2016) | 28.53 |
| *Our end-to-end system* | | | |
| Baseline | 21.89 | - | 31.00 |
| Sharpened Sigmoid | 21.64 | - | 29.73 |
| $G^2$-LSTM | 22.43 | - | **31.95** |

conditioned on previous words. A model is evaluated by the prediction perplexity: smaller the perplexity, better the prediction.

We followed the practice in (Merity et al., 2017) to set up the model architecture for LSTM: a stacked three-layer LSTM with drop-connect (Wan et al., 2013) on recurrent weights and a variant of averaged stochastic gradient descent (ASGD) (Polyak & Juditsky, 1992) for optimization. Our training code for $G^2$-LSTM was also based on the code released by Merity et al. (2017)[2]. We found the temperature $\tau$ used in $G^2$-LSTM is not very sensitive. We set the temperature to 0.9 and followed all configurations in Merity et al. (2017). We added neural cache model (Grave et al., 2016) on the top of our trained language model to further improve the perplexity.

### 4.1.2 MACHINE TRANSLATION

We used two datasets for experiments on neural machine translation (NMT): (1) IWSLT2014 German→English translation dataset (Cettolo et al., 2014), widely adopted in machine learning community (Bahdanau et al., 2016; Wiseman & Rush, 2016a; Ranzato et al., 2015). The train-

---

[2]https://github.com/salesforce/awd-lstm-lm

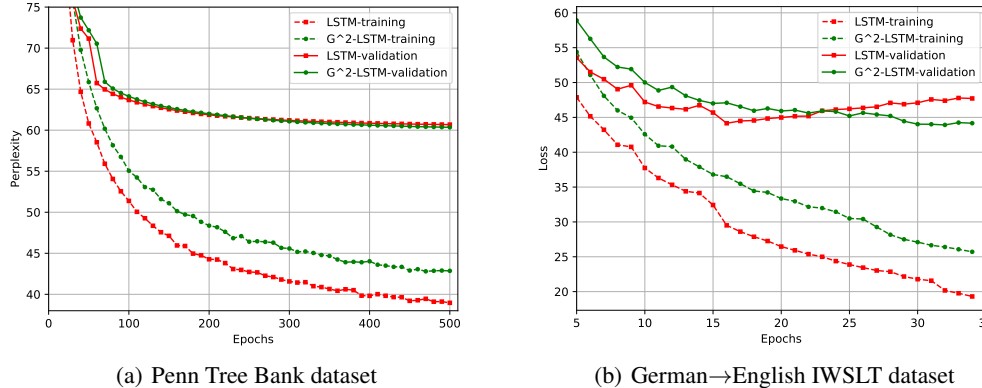

(a) Penn Tree Bank dataset        (b) German→English IWSLT dataset

Figure 2: Training/validation loss curves of language modeling and machine translation tasks.

ing/validation/test sets contains about $153k/7k/7k$ sentence pairs respectively, with words pre-processed into sub-word units using byte pair encoding (BPE) (Sennrich et al., 2016). We chose $25k$ most frequent sub-word units as vocabulary for both German and English. (2) English→German translation dataset in WMT'14, which is also commonly used as a benchmark task to evaluate d-ifferent NMT models (Bahdanau et al., 2014; Wu et al., 2016; Gehring et al., 2017). The training set contains 4.5M English→German sentence pairs, Newstest 2014 is used as the test set, and the concatenation of Newstest 2012 and Newstest2013 is used as the validation set. Similarly, BPE was used to form a vocabulary of most frequent $30k$ sub-word units for both language. In both datasets, we removed the sentences with more than $64$ sub-word units in training.

For German→English dataset, we adopted a stacked two-layer encoder-decoder framework. We set the size of word embedding and hidden state to 256. As amount of data in the English→German dataset is much larger, we adopted a stacked three-layer encoder-decoder framework and set the size of word embedding and hidden state to 512 and 1024 respectively. The first layer of the encoder was bi-directional. We also used dropout in training stacked LSTM as in (Zaremba et al., 2014), with dropout value determined via validation set performance. For both experiments, we set the temperature $\tau$ for $G^2$-LSTM to 0.9, which was the same as in the language model task. The mini-batch size was 32/64 for German→English/English→German respectively. All models were trained with AdaDelta (Zeiler, 2012) on one M40 GPU. Both gradient clipping norms were set to 2.0. We used tokenized case-sensitive BLEU (Papineni et al., 2002)[3] as evaluation measure. The beam size is set to 5 during the inference step.

## 4.2 EXPERIMENTAL RESULTS

The experimental results are shown in Table 1 and 2.

First, we compare our training method with two algorithms. For the first algorithm (we call it Baseline), we remove the Gumble-Softmax trick and train the model using standard optimization methods. For the second algorithm (we call it Sharpened Sigmoid), we use a sharpened sigmoid function as described in Section 3.2 by setting $\tau = 0.2$ and check whether such trick can bring better generalization. From the results, we can see that our learnt models are better than all base-line models. In language modeling task, we outperform the baseline algorithms for 0.7/1.1 points (1.2/1.4 points without continuous cache pointer) in terms of test perplexity. For machine trans-lation, we outperform the baselines for 0.95/2.22 and 0.54/0.79 points in terms of BLEU score for German→English and English→German dataset respectively. Note that the only difference between $G^2$-LSTM and the baselines is the training algorithm, while they adopt the same model structure. Thus, better results of $G^2$-LSTM demonstrate the effectiveness of our proposed training method.

---

[3]Calculated by the script at `https://github.com/moses-smt/mosesdecoder/blob/master/scripts/generic/multi-bleu.perl`

Second, training and validation loss curves of the baseline and $G^2$-LSTM are shown in Fig. 2 for the two small tasks. Both curves show that the gap between training and validation is effectively reduced using our algorithm. As shown in Fig. 2(b), the baseline LSTM achieves its lowest validation loss around the 18th epoch and begins to overfit after that, while the validation loss of $G^2$-LSTM still drops even in the 30th epoch. This clearly shows that $G^2$-LSTM generalizes better.

Third, we also list the performance of previous works in literature, which may adopt different model architectures or settings. For language modeling, we obtain the best performance as far as we know. For German→English translation, the two-layer stacked encoder-decoder model we learnt outperforms all previous works and achieves state-of-the-art performance. For English→German translation, our result is worse than GNMT (Wu et al., 2016) as they used a stacked eight-layer LSTM encoder-decoder model while we only used a three-layer one.

Table 3: Model compression results on Penn Tree Bank dataset

|  | Original | Round | Round & clip | SVD ($rank = 128$) | SVD ($rank = 64$) |
|---|---|---|---|---|---|
| Baseline | 52.8 | 53.2 (+0.4) | 53.6 (+0.8) | 56.6 (+3.8) | 65.5 (+12.7) |
| Sharpened Sigmoid | 53.2 | 53.5 (+0.3) | 53.6 (**+0.4**) | 54.6 (+1.4) | 60.0 (+6.8) |
| $G^2$-LSTM | **52.1** | **52.2 (+0.1)** | **52.8** (0.7) | **53.3 (+1.2)** | **56.0 (+3.9)** |

Table 4: Model compression results on IWSLT German→English dataset

|  | Original | Round | Round & clip | SVD ($rank = 32$) | SVD ($rank = 16$) |
|---|---|---|---|---|---|
| Baseline | 31.00 | 28.65 (-2.35) | 21.97 (-9.03) | 30.52 (-0.48) | 29.56 (-1.44) |
| Sharpened Sigmoid | 29.73 | 27.08 (-2.65) | 25.14 (-4.59) | 29.17 (-0.53) | 28.82 (-0.91) |
| $G^2$-LSTM | **31.95** | **31.44 (-0.51)** | **31.44 (-0.51)** | **31.62 (-0.33)** | **31.28 (-0.67)** |

Table 5: Model compression results on WMT English→German dataset

|  | Original | Round | Round & clip | SVD ($rank = 32$) | SVD ($rank = 16$) |
|---|---|---|---|---|---|
| Baseline | 21.89 | 16.22 (-5.67) | 16.03 (-5.86) | 21.15 (-0.74) | 19.99 (-1.90) |
| Sharpened Sigmoid | 21.64 | 16.85 (-4.79) | 16.72 (-4.92) | 20.98 (-0.66) | 19.87 (-1.77) |
| $G^2$-LSTM | **22.43** | **20.15 (-2.28)** | **20.29 (-2.14)** | **22.16 (-0.27)** | **21.84 (-0.51)** |

### 4.3 SENSITIVITY ANALYSIS

We conducted a set of experiments to test how sensitive our learnt models were if their gate parameters were compressed. We considered two ways of parameter compression.

**Low-Precision Compression** We compressed parameters in the input and forget gates to lower precision. Doing so the model can be compressed to a relatively small size. In particular, we applied round and clip operations to the parameters of the input and forget gates.

$$\text{round}_r(x) = \text{round}(x/r) * r \tag{17}$$
$$\text{clip}_c(x) = \text{clip}(x, -c, c). \tag{18}$$

We tested two settings of low-precision compression. In the first setting (named as *Round*), we rounded the parameters using Eqn (17). In this way, we reduced the support set of the parameters in the gates. In the second setting (named as *Round & Clip*), we further clipped the rounded value to a fixed range using Eqn (18) and thus restricted the number of different values. As the two tasks are far different, we set the round parameter $r = 0.2$ and the clip parameter $c = 0.4$ for the task of language modeling, and set $c = 1.0$ and $r = 0.5$ for neural machine translation. As a result, parameters of input gates and forget gates in language modeling can only take values from $(0.0, \pm0.2, \pm0.4)$, and $(0.0, \pm0.5, \pm1.0)$ for machine translation. More comprehensive results on different choices of hyperparameters can be found in Appendix A.

**Low-Rank Compression** We compressed parameter matrices of the input/forget gates to lower-rank matrices through single value decomposition. Doing so can reduce model size and lead to

faster matrix multiplication. Given that the hidden states of the task of language modeling were of much larger dimension than that of neural machine translation, we set $rank = 64/128$ for language modeling and $rank = 16/32$ for neural machine translation.

We summarize the results in Table 3-5. From Table 3, we can see that for language modeling both the baseline and our learnt model are quite robust to low-precision compression, but our model is much more robust and significantly outperforms the baseline with low-rank approximation. Even setting $rank = 64$ (roughly 12x compression rate of the gates), we still get 56.0 perplexity, while the perplexity of the baseline model increases from 52.8 to 65.5, i.e., becoming 24% worse. For machine translation, our proposed method is always better than the baseline model, no matter for low-precision or low-rank compression. Even if setting $rank = 16$ (roughly 8x/32x compression rate of the gates for German→English and English→German respectively), we still get roughly comparable translation accuracy to the baseline model with full parameters. All results show that the models trained with our proposed method are less sensitive to parameter compression.

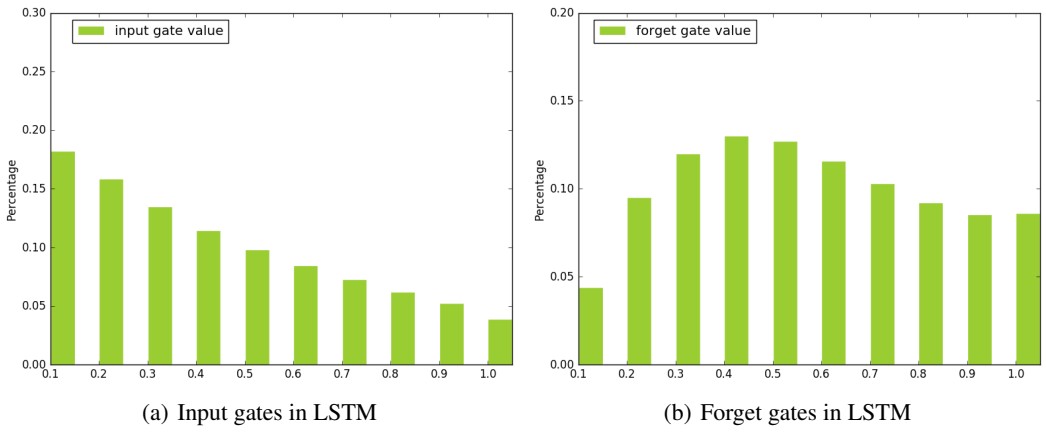

(a) Input gates in LSTM  (b) Forget gates in LSTM

Figure 3: Distributions of gate values in LSTM.

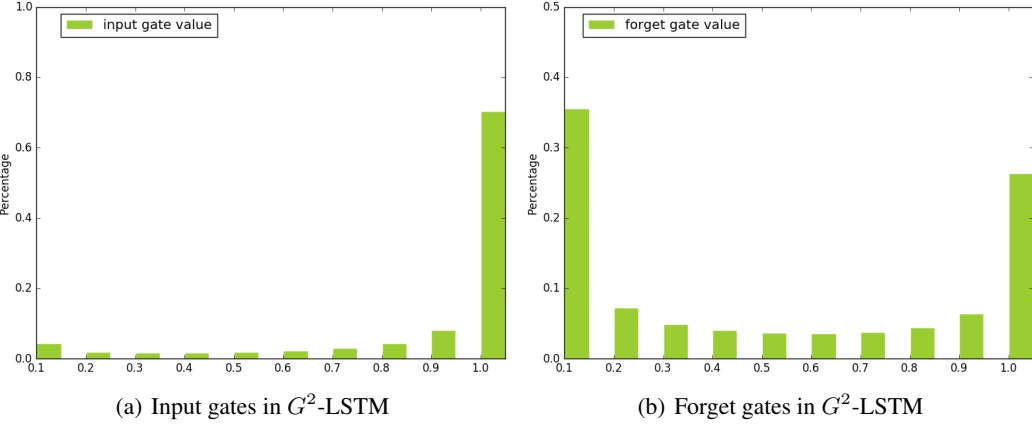

(a) Input gates in $G^2$-LSTM  (b) Forget gates in $G^2$-LSTM

Figure 4: Distributions of gate values in $G^2$-LSTM.

### 4.4 VISUALIZATION OF THE GATES

In addition to compare the final accuracy in previous two subsections, we further look inside the learnt models and check the values of the gates.

To well verify the effectiveness of our proposed $G^2$-LSTM, we did a set of experiments to show the values of gates we have learnt are near the boundary and are reasonable, based on the model learnt

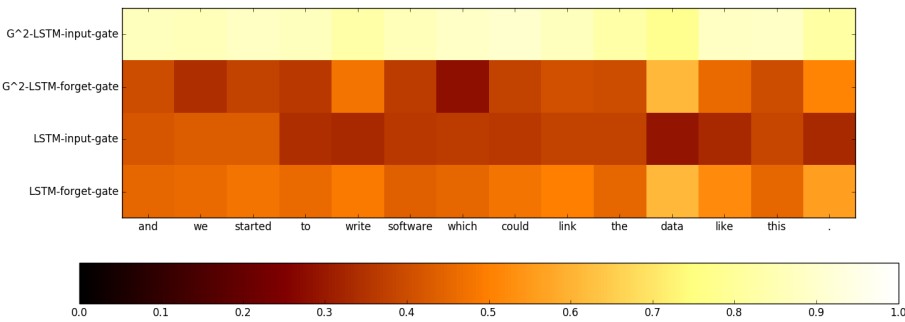

Figure 5: Visualization of gate values.

from German→English translation task. We show the value distribution of the gates trained using classic LSTM and $G^2$-LSTM. To achieve this, we sampled 10000 sentence pairs from the training set of German→English and fed them into the learnt models. We got the output value vectors of the input/forget gates in both the encoder and decoder. We recorded the value of each element in the output vectors and plotted the value distributions in Figure 3 and Figure 4.

From the figures, we can see that although both LSTM and $G^2$-LSTM work reasonably well in practice, the output values of the gates are very different. In LSTM, the distributions of the gate values are relatively uniform and have no clear concentration. In contrast, the values of the input gates of $G^2$-LSTM are concentrated in the region close to 1, which suggests that our learnt model tries to keep most information from the input words; the values of the forget gates are concentrated in the boundary regions (i.e., either the region close to 0 or the region close to 1). This observation shows that our training algorithm meets our expectation and successfully pushes the gates to 0/1.

Besides the overall distribution of gate values over a sampled set of training data, here we provide a case study for a sampled sentence. As it is hard to go deep into individual dimensions of a hidden state, we just calculated the average value of the output vector of the input and forget gate functions for each word. In particular, for each word, we focused on the average value of input/forget gate functions in the first layer and check whether the average is reasonable. We plot the heatmap of the English sentence part in Figure 5. More visualizations can be found in Appendix B. First, we can see that our $G^2$-LSTM does not drop information in the input gate function, since the average values are relatively large for all words. In contrast, the average values of the input gates of LSTM are sometimes small (less than 0.5), even for the meaningful word like "data". As those words are not included into LSTM, they cannot be effectively encoded and decoded, thus lead to bad translation result. Second, for $G^2$-LSTM, most of the words with small values for forget gates are function words (e.g., conjunctions and punctuations) or the boundaries in clauses. That is, our training algorithm indeed ensures the model to forget information on the boundaries inside the sentences, and reset the hidden states with new inputs.

## 5    CONCLUSION AND FUTURE WORK

In this paper, we have designed a new training algorithm for LSTM by leveraging the recently developed Gumbel-Softmax trick. Our training algorithm can push the values of the input and forget gates to 0 or 1, leading to robust LSTM models. Experiments on language modeling and machine translation have demonstrated the effectiveness of the proposed training algorithm.

We will explore following directions in the future. First, we have only tested with shallow LSTM models in this paper. We will apply our algorithm to deeper models (e.g., 8+ layers) and test on larger datasets. Second, we have considered the tasks of language modeling and machine translation. We will study more applications such as question answering and text summarization. Third, we are cleaning and refactoring the code and will release the training code to public soon.

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

## A  EXTRA EXPERIMENTS ON SENSITIVITY

We did an extra set of experiments on language modeling to show our model is less sensitive than the baseline model, no matter what the hyperparameters ($c, r$ in low-precision compression, $rank$ in low-rank compression) are. The results are shown in Table 6 and Table 7.

Table 6: Low precision compression results on Penn Tree Bank dataset

|  | Original | $c = 0.20, r = 0.10$ | $c = 0.40, r = 0.20$ | $c = 0.60, r = 0.30$ | $c = 0.80, r = 0.40$ |
|---|---|---|---|---|---|
| LSTM | 52.8 | 58.5 (+5.7) | 53.6 (+0.8) | 54.2 (+1.4) | 57.7 (+4.9) |
| Sharpened Sigmoid | 53.2 | 54.6 (+1.4) | 53.6 (**+0.4**) | 54.1 (**+0.9**) | 57.8 (+4.6) |
| $G^2$-LSTM | **52.1** | **54.5 (+2.4)** | **52.8** (+0.7) | **53.2** (+1.1) | **55.0 (+2.9)** |

Table 7: Low rank compression results on Penn Tree Bank dataset

|  | Original | $rank = 128$ | $rank = 64$ | $rank = 32$ | $rank = 16$ |
|---|---|---|---|---|---|
| LSTM | 52.8 | 56.6 (+3.8) | 65.5 (+12.7) | 83.1 (+30.3) | 111.6 (+58.8) |
| Sharpened Sigmoid | 53.2 | 54.6 (+1.4) | 60.0 (+6.8) | 72.8 (+19.6) | 100.9 (+47.7) |
| $G^2$-LSTM | **52.1** | **53.3 (+1.2)** | **56.0 (+3.9)** | **62.8 (+10.7)** | **75.9 (+23.8)** |

## B  EXAMPLES

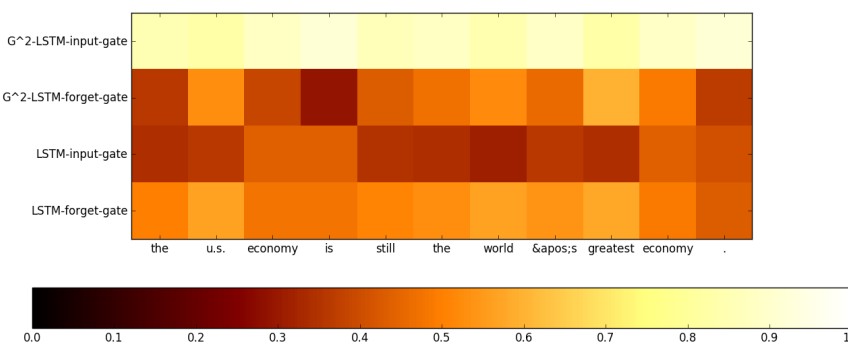

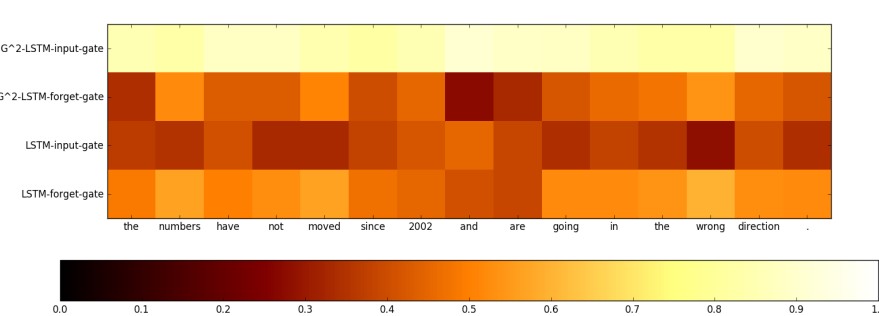

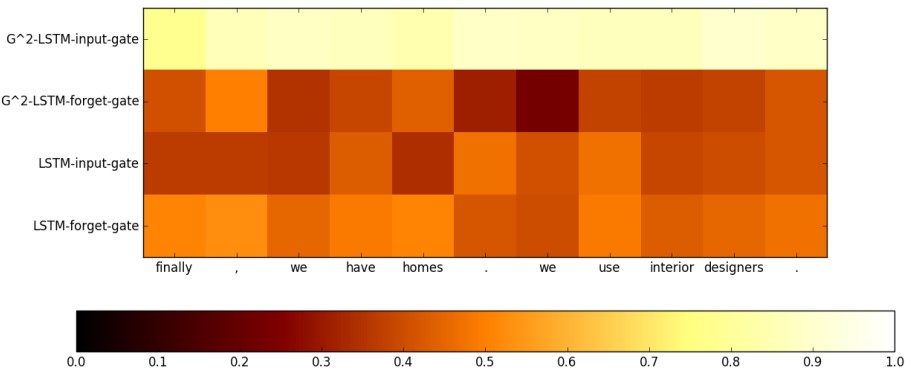

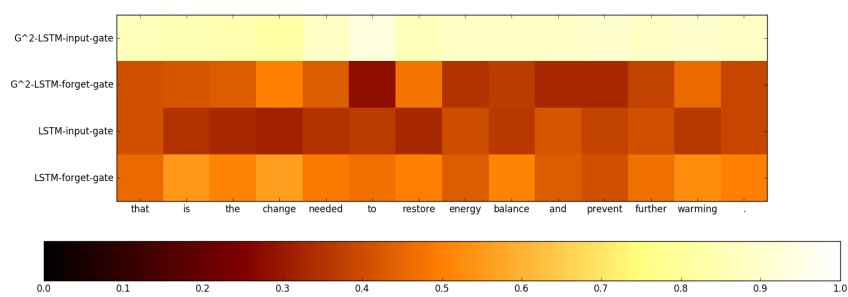

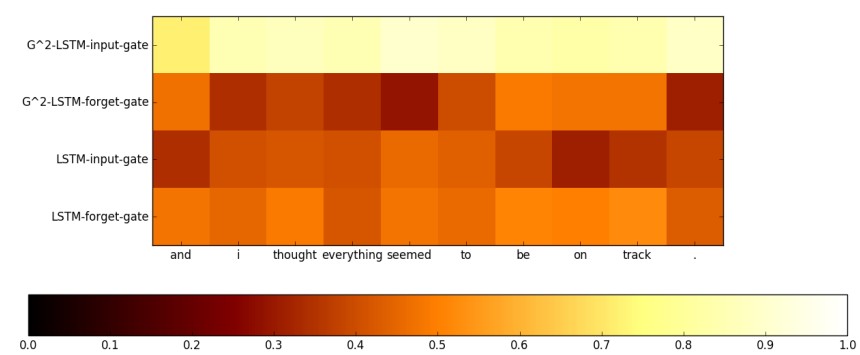

Figure 6: The gate value visualization in German→English task.

