# OpenReview forum: "Towards Binary-Valued Gates for Robust LSTM Training "
_ICLR.cc/2018/Conference — Reject_

### Official Review · AnonReviewer1 · 2017-11-26
**The technical novelty is limited and experiments do not show much benefits of the proposed model.**

**Rating:** 4
**Confidence:** 4

**Review:**

This paper aims to push the LSTM gates to be binary. To achieve this, the paper proposes to employ the recent Gumbel-Softmax trick to obtain end-to-end trainable categorical distribution (taking 0 or 1 value). The resulted G2-LSTM is applied for language model and machine translation in the experiments.

The novelty of this paper is limited. Just directly apply the Gumbel-Softmax trick.

The motivation is not explained clearly and convincingly. Why need to pursue binary gates? According to the paper, it may give better generalization performance. But there is no theoretical or experimental evidence provided by this paper to support this argument.

The results of the new G2-LSTM are not significantly better than baselines in the experiments.

---

> ### Author Response · Authors · 2017-12-27
> **We respectfully disagree with the comment about the novelty/experimental results of the paper.**
>
>
> [Regarding the experiment]
>
> We are afraid that the reviewer makes a wrong judgement to the performance results, our model is much better than the baseline on two tasks.
>
> For machine translation, we achieved the SOTA performance on German->English task and the improvement is significate (+ about 1 point) in the field of translation, not to mention that our model is much better than some other submissions https://openreview.net/forum?id=HktJec1RZ.
>
> For language model, by leveraging several tricks in literature, we significantly improve the performance from 77.4 to 52.1 (the best number as far as we know). This number is achieved without using any hyperparameter search method, we reported the detail in the paper.
>
> [Regarding the motivation]
>
> We have discussed in section 2.1 that there are a bunch of work empirically and theoretically studying the relationship between flat loss surface and generalization, not to mention that there are some continuous study and verification in ICLR 2018 submissions, e.g., https://openreview.net/forum?id=HkmaTz-0W . Thus our method is well motivated: by pushing the softmax operator towards its flat region will lead to better generalization.
>
> [Regarding the novelty of the paper]
>
> We are regretful to see the reviewer claims that there is little novelty in the paper. First, we are the first to apply Gumbel-softmax trick for robust training of LSTM by pushing the value of the gate to the boundary.  We empirically show that our method achieves better accuracy even achieves the SOTA performance in some tasks. Second, we show that by different low-precision/low-rank compressions, our model is even still comparable to the baseline models before compressions.

---

### Official Review · AnonReviewer3 · 2017-11-28
**Interesting, but not impressive**

**Rating:** 6
**Confidence:** 4

**Review:**

The paper argues for pushing the input and forget gate’s output toward 0 or 1, i.e., the LSTM tends to reside in flat region of surface loss, which is likely to generalize well. To achieve that, the sigmoid function in the original LSTM is replaced by a function G that is continuous and differentiable with respect to the parameters (by applying the Gumbel-Softmax trick). As a result, the model is still differentiable while the output gate is approximately binarized.

Pros:
-	The paper is clearly written
-	The method is new and somehow theoretically guaranteed by the proof of the Proposition 1
-	The experiments are clearly explained with detailed configurations
-	The performance of the method in the model compression task is promising

Cons:
-	The “simple deduction” which states that pushing the gate values toward 0 or 1 correspond to the region of the overall loss surface may need more theoretical analysis
-	It is confusing whether the output of the gate is sampled based on or computed directly by the function G
-	The experiments lack many recent baselines on the same dataset (Penn Treebank: Melis et al. (2017) – On the State of the Art of Evaluation in Neural Language Models; WMT: Ashish et.al. (2017) – Attention Is All You Need)
-	The experiment’s result is only slightly better than the baseline’s
-	To be more persuasive, the author should include in the baselines other method that can “binerize” the gate values such as the one sharpening the sigmoid function.


In short, this work is worth a read. Although the experimental results are not quite persuasive, the method is nice and promising.

---

> ### Author Response · Authors · 2017-12-27
> **Thanks for the relevant comments. We have improved PTB results according to the suggestions.**
>
>
> [Regarding the computation of function G]
>
> During training, the output of the gate is computed directly by function G, while the function G contains some random noise U.
>
> [Regarding the sharpened sigmoid function experiment]
>
> Thanks for figure this out. First, we want to point out that theoretically it doesn’t help: Simply consider function f_{W,b}(x) =sigmoid((Wx+b)/tau), where tau is the temperature, it is computationally equivalent to f_{W’,b’}(x) =sigmoid(W’x+b’) by setting W’=W/tau and b’ = b/tau. Then using a small temperature is equivalent to rescale the initial parameter as well as gradient to a larger range. Usually, setting an initial point in a larger range with a larger learning rate will harm the optimization process.
>
> We also did a set of experiments and updated the paper to show it doesn’t help in practice.
>
> [Regarding the significance of experimental results]
>
> For machine translation, we achieved the SOTA performance on German->English task and the improvement is significate (+ about 1 point) in the field of translation, not to mention that our model is much better than some other submissions https://openreview.net/forum?id=HktJec1RZ. For English->German task, we noticed that “Attention is all you need” is the state of the art but it is not LSTM-based; thus we didn’t list that result in the paper.
>
> For language model, thanks for the reference, we have studied the papers. By leveraging several tricks in literature, we significantly improve the performance from 77.4 to 52.1 (the best number as far as we know) without using any hyperparameter search method, we reported the detail in the paper.

---

### Official Review · AnonReviewer2 · 2017-11-28

**Rating:** 6
**Confidence:** 3

**Review:**

This paper propose a new "gate" function for LSTM to enable the values of the gates towards 0 or 1. The motivation behind is a  flat region of the loss surface is likely to generalize well. It shows the experimental results are comparable or better than vanilla LSTM and much more robust to low-precision approximation and low-rank approximation.

In section 3.2, the paper claimed using a smaller temperature cannot guarantee the outputs to be close to the boundary. Is there any experimental evidence to show it's not working? It also claimed pushing output gate to 0/1 will drop the performance. It actually quite interesting because there are bunch of paper claimed output gate is not important for language modeling, e.g. https://openreview.net/pdf?id=HJOQ7MgAW .

In the sensitive analysis, what if apply rounding / low-rank for all the parameters?

How was this approach compare to binarynet https://arxiv.org/abs/1602.02830 ? Applying the same idea, but only for forget gate/ input gate. Also, can we apply this idea to the binarynet?

Overall, I think it's an interesting paper but I feel it should compare with some simple baseline to binarized the gate function.

Updates: Thanks a lot for all the clarification. It do improve the paper quality but I'm still thinking it's higher than "6" but lower than "7". To me, improve ppl from "52.8" to "52.1" isn't very significant. For WMT, it improve on DE->EN but not for EN->DE (although it improve both for the author's own baseline). So I'm not fully convinced this approach could improve the generalization. But I feel this work can have many other applications such as "binarynet".

---

> ### Author Response · Authors · 2017-12-27
> **Thanks for the relevant comments. Here are the responses to the questions.**
>
> [Regarding the small temperature experiment]
>
> Thanks for figure this out. First, we want to point out that theoretically it doesn’t help: Simply consider function f_{W,b}(x) =sigmoid((Wx+b)/tau), where tau is the temperature, it is computationally equivalent to f_{W’,b’}(x) =sigmoid(W’x+b’) by setting W’=W/tau and b’ = b/tau. Then using a small temperature is equivalent to rescale the initial parameter as well as gradient to a larger range. Usually, setting an initial point in a larger range with a larger learning rate will harm the optimization process.
>
> We also did a set of experiments and updated the paper to show it doesn’t help in practice.
>
> [Regarding the binary net]
>
> Despite the different between the model structure (gate-based LSTM v.s. CNN), the main difference is that we regularize the output of the activation of the gates to binary value only, but not to regularize the weights. One should notice that the accuracy of Binary Net is usually much worse than the baseline model. However, we show that (1) Our models generalize well among different tasks. (2) The accuracy of the models after low-rank/low-precision compression using our method is competitive to (or even better than) the baseline. Besides, our techniques can also be applied to binarynet training.
>
> [Regarding apply rounding / low-rank for all the parameters]
>
> We will do the experiment but as our proposed method is focusing on LSTM unit. We are not sure whether the performance will drop a lot when we apply rounding/low-rank to embedding and attention.

---

### Author Response · Authors · 2018-01-04
**Revision to the paper (to all reviewers) : updates on experimental results**

Thanks all reviewers for their valuable comments, we updated a new version of the paper by including the following results:

1. We make discussion about the sharpening sigmoid method proposed by the reviewers, and add the algorithm as one of the baselines in the experiments. The experimental results still show that our proposed method achieves the best performance in all tasks.

2. We update the experimental results on language modelling task which achieves the best performance (52.1) as far as we know without using any hyperparameter search method.

---

### Decision · Program_Chairs · 2018-01-29
**ICLR 2018 Conference Acceptance Decision**

**Decision:**

Reject

**Comment:**

This paper proposes training binary-values LSTMs for NLP using the Gumbel-softmax reparameterization.  The motivation is that this will generalize better, and this is demonstrated in a couple of instances.

However, it's not clear how cherry-picked the examples are, since the training loss wasn't reported for most experiments.  And, if the motivation is better generalization, it's not clear why we would use this particular setup.